# Correlation between two distant quasiparticles in separate superconducting islands mediated by a single spin

Juan Carlos Estrada Saldaña [1], Alexandros Vekris[1,2], Luka Pavešič [3,4], Rok Žitko [3,4] ✉, Kasper Grove-Rasmussen[1] & Jesper Nygård [1] ✉

Controlled coupling between distant particles is a key requirement for the implementation of quantum information technologies. A promising platform are hybrid systems of semiconducting quantum dots coupled to super-conducting islands, where the tunability of the dots is combined with the macroscopic coherence of the islands to produce states with non-local correlations, e.g. in Cooper pair splitters. Electrons in hybrid quantum dots are typically not amenable to long-distance spin alignment as they tend to be screened into a localized singlet state by bound superconducting quasiparticles. However, two quasiparticles coming from different superconductors can overscreen the quantum dot into a doublet state, leading to ferromagnetic correlations between the superconducting islands. We present experimental evidence of a stabilized overscreened state, implying correlated quasiparticles over a micrometer distance. We propose alternating chains of quantum dots and superconducting islands as a novel platform for controllable large-scale spin coupling.

Chains of quantum dots (QD) and superconducting islands (SI) can be fabricated by appropriately patterning and locally gating superconductor-semiconductor hybrid nanowires. These systems are excellent for exploring non-local properties of discrete states in superconducting gaps (subgap states)[1], qubits[2–4], and non-local processes[5–11] for topological[12,13] and non-topological chains[14]. The subgap states in QD-SI heterostructures are induced by the spin exchange (Kondo) interaction binding a Bogoliubov quasiparticle and the QD magnetic moment into a singlet state[15–23]. This is a super-conducting realisation of the Kondo effect – screening of a localized impurity spin by itinerant particles from a bath. With multiple channels coupled to the dot, more exotic spin states can arise. In the normal-state two-channel Kondo effect, the ground state exhibits a phenom-enon called overscreening[24,25]: electrons from two separate channels compete to screen the spin, leading to a frustrated doublet state where the impurity is completely screened, but a many-body spin-1/2

remains, smeared across the system. This is an unstable fixed point of the renormalization group flow which only exists for symmetric cou-pling to both leads. In the case of an asymmetry, the system flows towards the state where the screening comes completely from the more strongly coupled channel[25]. For this reason, difficult fine-tuning is required to demonstrate the overscreened state in QD devices[26,27].

Overscreening is energetically disfavored in the superconducting (SC) case as it requires the presence of two additional finite-energy quasiparticles. However, SIs with large charging energy can be tuned into an odd-occupancy regime where each SI contains one lone quasiparticle[22,28–31] and here a superconducting version of over-screening emerges. In superconducting systems, the Kondo renor-malization process is cut off at the energy scale of the gap[32], and overscreening is expected to exist even if the couplings to both SIs are not strictly the same, i.e., the device does not need to be perfectly mirror (left-right) symmetric. The theoretical model that we present

[1]Center for Quantum Devices, Niels Bohr Institute, University of Copenhagen, 2100 Copenhagen, Denmark. [2]Sino-Danish College (SDC), University of Chinese Academy of Sciences, Beijing, China. [3]Jožef Stefan Institute, Jamova 39, SI-1000 Ljubljana, Slovenia. [4]Faculty of Mathematics and Physics, University of Ljubljana, Jadranska 19, SI-1000 Ljubljana, Slovenia. ✉e-mail: rok.zitko@ijs.si; nygard@nbi.ku.dk

indeed anticipates such a state and furthermore shows that the screening of the QD comes from two quasiparticles: one occupying an orbital that spans the two SIs and couples to the QD ("bonding" orbital), while a second decoupled orthogonal orbital carries the residual free spin ("antibonding" orbital). The latter results in ferromagnetic correlations between the SIs. In this paper, we show signatures of such a superconducting overscreened state.

## Results

### Modelling the ground states of a chain

The system under study is an SI-QD-SI chain made from an InAs semiconductor nanowire, Fig. 1a, that we will describe in more detail below. Due to the complexity of the system, we first present a simplified model[30,33] as a theoretical framework for discussing the experimental results. It is schematically represented in Fig. 1b and involves two SIs modelled by the Richardson model[29] as sets of energy levels experiencing pairing attraction. $\Delta$ is the resulting SC gap[28,29,31,34,35] and $E_c$ is the charging energy. The SI filling is controlled by a gate voltage, favoring the occupancy of $n_0$ electrons. The QD is modelled with the Anderson impurity model[36] with on-site repulsion $U$[37,38]. The QD is coupled to the SIs through single-electron hopping $v$, quantified by the hybridization $\Gamma \propto v^2/U$. See Methods for further details.

Figure 1e shows the low-energy spectra vs. $\Gamma$ at odd SI filling $n_0$. The case of $E_c < \Delta$ (top) is reminiscent of the standard superconducting Kondo effect[39]. At small $\Gamma$ the ground state is a decoupled doublet (blue), where the two SIs contain condensates of Cooper pairs and the free spin is localized at the QD. With increasing $\Gamma$ two discrete states decouple from the continuum of singlet excitations (red), the lower being a gerade superposition of a single quasiparticle in each SI completely screening the QD spin[17,33,39]. Due to its symmetry we denote

this state the LR singlet; it is sketched in Fig. 1d. The doublet continuum is composed of excited states with a broken Cooper pair, with an energy cost $2\Delta$, resulting in two Bogoliubov quasiparticles, free to redistribute among the two SIs. For odd $n_0$, each occupies one SI to minimize the charging energy penalty, recovering $2E_c$. The gap in the doublet sector for $\Gamma \to 0$ is thus $2(\Delta - E_c)$. The doublet states in such systems are typically understood to contain a free or at most partially screened QD spin[40], with only the singlet subgap states exhibiting strong screening of the QD spin. However, the charge configuration of the excited states - namely the presence of unpaired quasiparticles - enables QD spin screening in the doublet manifold as well. This further decreases the doublet gap with increasing $\Gamma$, but not to the point of completely closing it.

When $E_c > \Delta$ (Fig. 1(e) bottom), it becomes energetically favourable to break a Cooper pair in the doublet ground state and distribute the resulting quasiparticles across the two SIs, and thus have odd occupation in both. Increasing $\Gamma$ in this regime produces a discrete in-gap doublet state - the ground state becomes the overscreened (OS) doublet, sketched in Fig. 1c. The OS is a complex many body state involving two quasiparticles in the two SIs and the electron on the QD.

Importantly, we find a striking difference between the $E_c < \Delta$ and $E_c > \Delta$ cases. For $E_c > \Delta$, we find that the ground states in the singlet (LR) and doublet (OS) sector gain approximately the same binding energy $E_B$, which leads to the saturation of the OS → LR excitation energy (black arrows) at large coupling. This implies that the screening mechanism in the OS doublet is very similar to that in the LR singlet state (associated with a quasiparticle in the "bonding" orbital), with the remaining spin-carrying quasiparticle occupying the orthogonal ungerade ("antibonding) orbital decoupled from the QD[24,26,27,32,41,42]. Importantly, we find that in the $E_c > \Delta$ regime the doublet → singlet

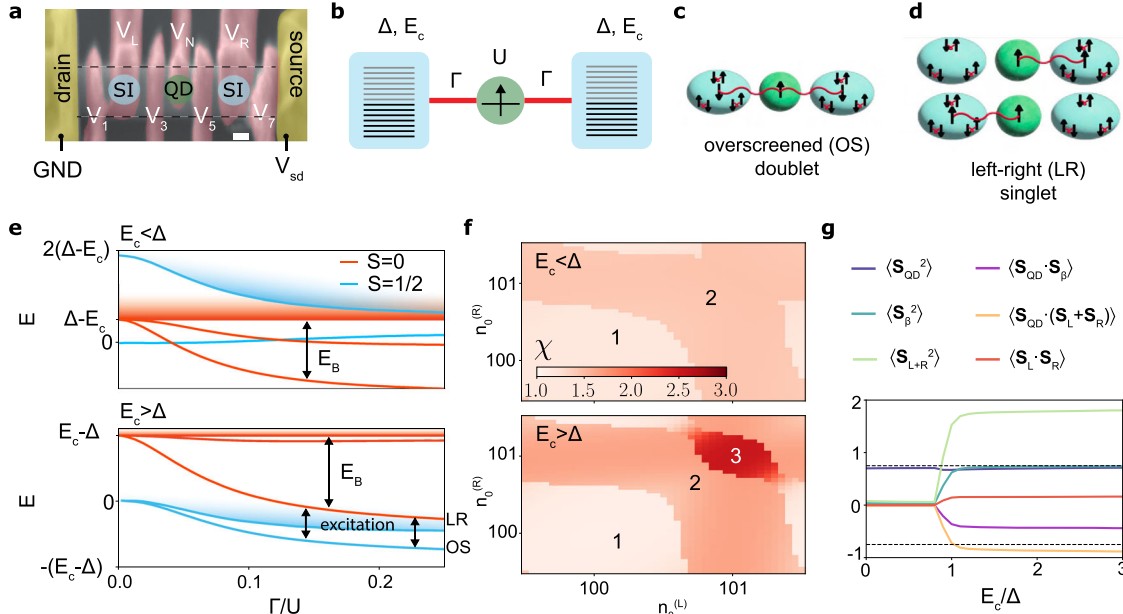

**Fig. 1 | Ground-state properties for $E_c < \Delta$ and $E_c > \Delta$: model calculations.**
**a** Device used to realize the three-element chain consisting of a QD coupled to two SIs on an InAs nanowire (dashed). Scale bar is 100nm. **b** Model sketch. Two SIs with equal superconducting gap $\Delta$ and charging energy $E_c$, described by the Richardson model, are coupled to a QD, modelled by the single-impurity Anderson model. The SI occupation is enforced with $E_c(\hat{n}_{SC} - n_0)^2$ terms. See Methods for details. **c** Sketch of the overscreened doublet state, the doublet GS at odd tuning. **d** Sketch of the left-right singlet state, the singlet GS at even tuning. The GS is an equally weighted linear superpositions of the two states shown. The squiggle represents spin coupling that results in some degree of entanglement. **e** Energy spectrum vs. hybridisation $\Gamma$ at odd tuning ($n_0$ = odd) for (top) $E_c/\Delta = 0.5$ and (bottom) $E_c/\Delta = 2$,

both $U/\Delta = 3$. Red: singlets with even total charge, blue: doublets with odd total charge. The continuum is indicated by shading. To remove the overall linear decrease of all energies with $\Gamma$, the energy of the singlet continuum is subtracted, and the doublet ground state at $\Gamma \to 0$ set to zero energy. **f** Number of local magnetic moments in the GS, quantified by $\chi = \frac{4}{3}(\chi_{QD} + \chi_L + \chi_R)$, in the plane of left and right SI filling ($n_0^L$, $n_0^R$), for (top) $E_c/\Delta = 0.5$ and (bottom) $E_c/\Delta = 2$. $U = 5\Delta$, $\Gamma = 0.1U$ for both. The number of local magnetic moments is computed from the spin-spin correlations $\chi_\beta = \sum_{i,j \in \beta} \mathbf{S}_i \cdot \mathbf{S}_j$. For a free spin $\chi_\beta = \frac{3}{4}$, thus the total is multiplied by $\frac{4}{3}$ to obtain the number of LMs. **g** Partial sums of the spin-spin correlation matrix in the doublet ground state for $n_0$ = odd as a function of $E_c$. Here $\mathbf{S}_\beta = \sum_{i \in \beta} \mathbf{S}_i$. $L + R$ refers to the union of the left and right SI. Parameters are $U/\Delta = 3$, $\Gamma/U = 0.1$.

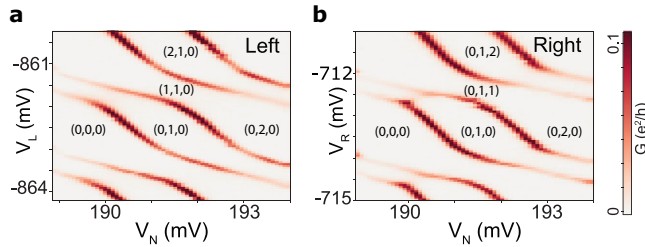

**Fig. 2 | Device tunability allowing high left-right symmetry.** Zero-bias conductance $G$ versus gate voltages $V_N$, and (**a**) $V_L$ or (**b**) $V_R$. The resemblance in the conductance patterns in the two diagrams reflects the high degree of left-right symmetry of the device parameters in this gate configuration, with the differences in $(E_c/\Delta)^*$ and binding energy $E_B$ being 8% and 14%. Here $(E_c/\Delta)^*$ and $E_B$ are parameters gauged from the charging diagrams as presented in Supplementary information Fig. S3, while the relation to theoretical parameters $E_c/\Delta$ and $\Gamma$ is shown in Supplementary information Fig. S4. The numbers indicate the occupancy of left island, dot, and right island.

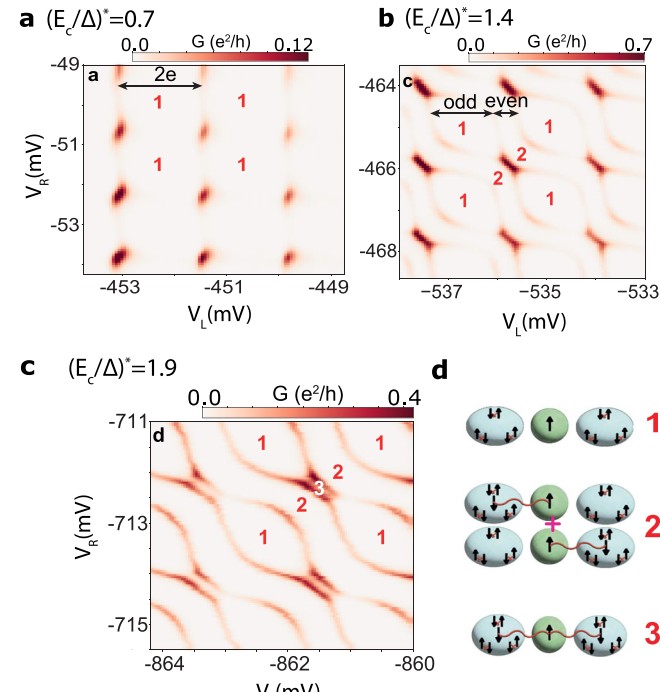

**Fig. 3 | Emergence of overscreened doublet regions with 3 LMs in charging diagrams.** **a**-**c** Zero-bias conductance $G$ versus the two (left/right) SI gate voltages $V_L$, $V_R$ for three values of $(E_c/\Delta)^*$. Numbers in the diagram indicate the total number of LMs in the device, supported by calculations in Fig. 1; schematic diagrams present the character of these states (**d**).

phase transition, ubiquitous in this class of devices for low values of $E_c$[17,30,43], does not occur at any value of $\Gamma$.

The eigenstates of the system are complex many-body superpositions of different distributions of charge and alignments of their spins, with the dominant part changing with parameters. For example, the doublet-singlet excitation energy actually slightly decreases at small coupling, because the doublet GS in this regime is a superposition of the decoupled QD and the OS state; the crossover to a pure OS state occurs only at larger coupling $\Gamma$. When the OS state becomes the dominant component the excitation energy indeed saturates. The multitude of most relevant states in the system are shown in Supplementary information Fig. S1.

The presence of two superconducting quasiparticles is a distinctive feature of the OS state. We gauge it by plotting the number of local moments (LMs) $\chi = \frac{4}{3}\sum_{\beta=QD,L,R}\langle S_\beta^2\rangle$ in the plane of the two SI fillings $(n_0^{(L)}, n_0^{(R)})$ in Fig. 1f. For the decoupled doublet we expect $\chi \approx 1$ coming from the spin in the QD, while the subgap singlets with a single quasiparticle have $\chi \approx 2$. For $E_c > \Delta$ and in the vicinity of odd $n_0$s, a new lobe emerges with $\chi \approx 3$. This is the OS ground state. We use the emergence of the OS lobe at odd SI filling as an experimental signature of the presence of overscreening in our device.

The spin properties of the ground states in the singlet and doublet sectors evolve very differently with increasing $E_c$. The LR singlet (not shown) hardly changes. The QD carries a LM, so that $\langle S_{QD}^2\rangle = 3/4$. This is exactly matched by the total magnetic moment present in *both* SIs, $\langle S_{L+R}^2\rangle$.

In the doublet sector, shown in Fig. 1g, we find a striking transition between the state with a lone LM in the QD at $E_c < \Delta$ and rich spin properties of the OS state for $E_c > \Delta$ (see also Supplementary information Fig. S2). In the OS state, two further LMs emerge, so that $\langle S_L^2\rangle = \langle S_R^2\rangle \approx \frac{3}{4}$ (cyan curve in Fig. 1g). As in the singlet case, the screening comes from both SIs. However, importantly, the spin states of the two SIs are correlated. We find inter-island spin correlations $\langle S_L \cdot S_R\rangle \sim \frac{1}{4}$, while $\langle S_{L+R}^2\rangle$ approaches 2 (red and green curves in Fig. 1g, respectively). Both are signs of ferromagnetic correlation, implying that the spins of the quasiparticles collaboratively screening the QD are aligned.

It is worth commenting on the differences between Kondo-type Hamiltonians with no charge transfer processes, and more realistic Anderson-type Hamiltonians that more adequately model quantum dots. Even in the presence of charging energy on SIs, the Anderson model does not behave in the same way as the Kondo model. This is due to the hybridisation, which leads to the formation of bonding and antibonding orbitals, as discussed above. For this reason, the physics of the overscreened state in the Kondo case[32] is different from that of

the overscreened state in the Anderson case discussed in this work. In particular, there is no self-dual point and no universality of the energy of the subgap state.

## State emerges in experiment

The presence of overscreened doublet states was explored in an SI-QD-SI device, Fig. 1a. In this system both SIs are adjusted to have approximately identical properties, which is possible because $E_{cL}/\Delta_L$ and $E_{cR}/\Delta_R$ can be individually tuned by coarse changes in gate voltages $V_L$ and $V_R$. Furthermore, for $E_{cL} > \Delta_L$ and $E_{cR} > \Delta_R$, the occupations of the SIs can be accurately tuned by further fine adjustments of $V_L$ and $V_R$. The QD occupancy is tuned with the top-gate voltage $V_N$, while the binding energies $E_{BL}$, $E_{BR}$ to the two Al SIs are controlled by $V_3$ and $V_5$ (See Methods for details). Standard lock-in techniques are used to obtain the differential conductance $G$, from which we extract excitation energies. We tune the device to left-right symmetric $E_c/\Delta$ and $E_B$ by comparing pairs of zero-bias $G$ diagrams, of which an example is shown in Fig. 2. The high symmetry that we are able to achieve relies on high device tunability and on designing the SIs to be nominally identical by crystal growth and lithography, an advantage over gate-defined QD chains[44].

The competition between various ground states is experimentally investigated by sweeping the gate voltages that control the number of LMs and their distribution within the device. Phase diagrams in the $(V_L, V_R)$ plane are shown in Fig. 3 for a range of increasing $E_c/\Delta$, with superposed numbers indicating the number of unpaired spins in the device. $V_N$ is tuned so that the QD is occupied by a single electron. The initial $2e$ charging regime of the islands for low $E_c/\Delta$ (Fig. 3a; only 1 LM on the QD) is broken for higher charging energy (Fig. 3b). When $E_c > \Delta$, and $V_L$, $V_R$ both fine tuned for odd occupation, a new doublet lobe with three spins emerges (Fig. 3c) as the overscreened state becomes the ground state, in agreement with Fig. 1f. The appearance of this new

ground state for increased $E_c$ (Fig. 3c), as predicted by theory, is a main result of this study. Additional data allows us to trace the emerging overscreened state; e.g. in bias spectroscopy the OS states are also revealed, see Supplementary information Fig. S5.

## Spectroscopy in field

Further evidence for the presence of the OS state is obtained by using a magnetic field, $B$, to polarize the magnetic moments. This leads to qualitative changes in the spectrum as the triplet and quadruplet states come into play (see Supplementary information Fig. S1.) Bias spectroscopy plots for weak and strong binding at two $B$ values in Fig. 4a-d illustrate the extraction of the excitation energies that are presented in Fig. 4e, accompanied by the model calculations in Fig. 4f to guide the interpretation. The applied magnetic field is parallel to the nanowire axis and much weaker than the in-plane critical field of the superconducting shell.

In the 1 LM regime (SIs tuned to even occupation, QD to odd), at zero field the GS is a doublet with the LM in the QD (Fig. 4f). It splits in the presence of $B$, while the subgap singlet remains unperturbed. The excitation energy thus increases proportionally to $g_N/2$, $g_N$ being the g-factor of the QD (Fig. 4e). Simultaneously, a triplet state with one LM in the QD and the other distributed symmetrically across the two SIs descends in energy with a rate proportional to $g_N/2 + (g_L + g_R)/4$, where $g_L$ and $g_R$ are g-factors of the left and right SI. Since the triplet decreases in energy at a higher rate than the doublet, the excitation energy starts to decrease at this point (see red bullet). The model predicts another change of slope when the $S = 3/2$ quadruplet becomes the GS, but this regime is not reached in the experiment. The comparison with theory is made for a dataset corresponding to weak binding, so that local moments in all subsystems are better defined.

In the 3 LM regime (SIs and QD tuned to odd occupation), at zero field the OS doublet and the LR singlet are separated from the continuum of higher-spin excitations in their corresponding subspaces by the same binding energy $E_B$ (Fig. 4f). Furthermore, in each subspace the excitation energy decreases with $B$ at the same rate relative to the respective GS, meaning that the singlet-triplet and doublet-quadruplet crossings (green circles) occur at roughly the same $B$. The excitation energy thus has a constant linear dependence versus $B$ despite the level crossing. The slope is proportional to $(g_L + g_R)/4$, owing to the odd-occupancy states differing by an additional spin-polarised quasiparticle spread over both SIs (Fig. 4e). The comparison with theory is here made for a strong-binding dataset in order to stabilize the overscreened state. The slight disagreement of the experiment with this simple interpretation is mostly due to unequal g factors. While in the model we assume $g_L = g_N = g_R$, we experimentally obtain $g_L = 8.7$, $g_N = 17$, $g_R = 5.9$ for the weak-binding dataset, and $g_L = 8.8$, $g_N = 20$, $g_R = 5.7$ for the strong-binding dataset. These are measured by loading a single LM to the relevant device component by tuning appropriate gate voltages, see Supplementary information Fig. S7. Additional data is shown in Supplementary information Fig. S8.

## Discussion

In conclusion, we presented experimental evidence for the existence of a superconducting overscreened subgap state in a SI-QD-SI chain. This is a doublet ground state in which the spin residing on the QD is completely screened in the same way as in the singlet subgap states, adding to the body of evidence of screening in the doublet state[40].

Furthermore, the overscreened state is predicted to exhibit long-distance spin correlations between the quasiparticles in the SIs. These correlations occur on the micrometer scale (device size), in contrast to

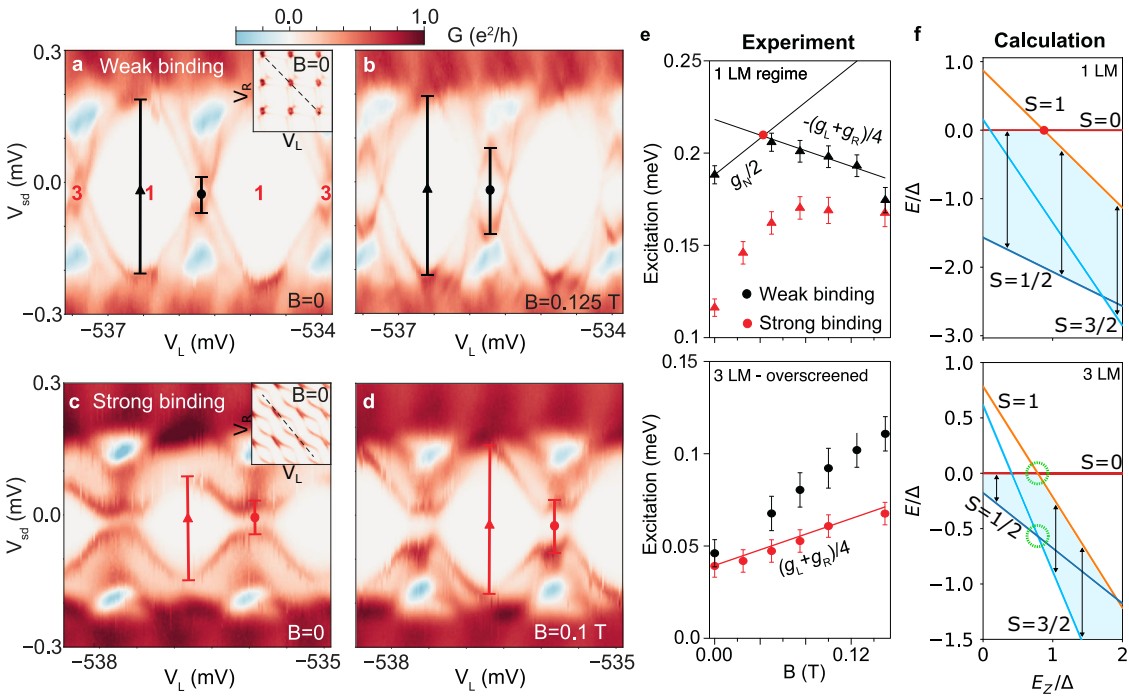

**Fig. 4 | Polarizing two bound Bogoliubov quasiparticles with magnetic field.** Bias spectra for approximately left-right symmetric parameters, with (**a**, **b**) $E_B/2E_c = 0.04$ (weak binding), $E_c/\Delta = 1.45$ and (**c**, **d**) $E_B/2E_c = 0.32$ (strong binding), $E_c/\Delta = 1.65$, recorded at different $B$ indicated on each plot. The gate sweep, indicated by a dashed line in the inset stability diagrams in (a, c) for $B = 0$, alternates the total number of LMs between 1 and 3, while keeping a LM fixed in the QD. At $B = 0$, the GS is a doublet and the first excited state a singlet. These states are overran by higher-spin states at larger $B$. **e** $B$ dependence of excitation energy for weak (black symbols) and strong (red symbols) binding, in the regime with either 1 LM (top plot) or

3 LM (bottom plot); in the latter case the GS is the OS state. The lines indicate theory estimates, red dot marks the crossing point (see text for details). **f** Calculation of the spectrum versus the Zeeman energy of the QD, $E_Z$, assuming equal g-factors in all parts of the device, in the regime with either 1 LM (top plot) or 3 LM (bottom plot). States are labeled with total spin $S$. Black arrows indicate the energy difference between the ground state in the sector with odd and even number of particles; this is the excitation energy shown in (e). Green circles in the bottom plot indicate the singlet-triplet and doublet-quadruplet crossings. $U/\Delta = 4$, $E_c/\Delta = 1.5$ and $\Gamma/U = 0.3$.

the nanometer scale of YSR chains of magnetic adatoms on $E_c = 0$ superconducting substrates[45]. We propose to utilize them as a source of long-distance correlations in a condensed-matter system. The ground state of an extended chain of alternating SIs and QDs, tuned to odd occupancy and $E_c > \Delta$, is a version of the OS state replicated across all SIs. Preliminary calculations show that in this case, the ferromagnetic correlations stabilized by large $E_c$ extend throughout the whole chain, see Supplementary information Fig. S9.

In this sense, the SI-QD-SI system considered here is a possible building block for a longer device, and understanding it is an important first step in the pursuit of this class of large-scale quantum simulators. For example, a longer odd-length chain can be used to demonstrate the self-similarity of the two-channel Kondo state on a tight-binding chain[24], where the central extended doublet is recursively overscreened. The length dependence of the correlations can be investigated by unloading LMs from the elements (QD or SI) of the long chain, effectively shortening its length. Even-length chains of QD-SI singlet dimers should instead lead to quasi-long-range antiferromagnetic correlations between the end unpaired elements[46].

The chain can be mapped to other well-known models by setting its parameters to various special limits. For example, for $\Delta = 0$ it maps to the Hubbard chain (triple QD for 3 sites) and, for weak hopping (or large $U$), to the Heisenberg chain. For $E_{BL} \neq E_{BR}$, it realizes the interacting Su-Schrieffer-Heeger model[47], and for $E_c = 0$, $\Delta = 0$ it simulates the Kondo necklace[48]. Extension to two-dimensional lattices is possible by using nanowire networks[49–51], enabling the pursuit of topological spin liquids[52].

## Methods

### Model and calculations

For calculations in Fig. 1, we describe the QD as a single nondegenerate impurity level, as in the single-impurity Anderson model[36]. The SIs are described by the Richardson model, as two sets of equidistant energy levels that represent time-reversal-conjugate pairs in the momentum/orbital space[29]. These are coupled all-to-all by the pairing interaction. This step beyond the BCS mean-field approximation allows for particle number conservation and is required to accurately describe even-odd occupancy effects of the SI with large charging energy $E_c$. The QD is coupled to all levels of both SIs with the hybridisation terms. The Hamiltonian is

$$H = H_{QD} + \sum_{\beta = L,R} \left( H_{SC}^{(\beta)} + H_{hyb}^{(\beta)} \right), \tag{1}$$

where

$$H_{QD} = \varepsilon_{QD} \hat{n}_{QD} + U \hat{n}_{QD,\uparrow} \hat{n}_{QD,\downarrow} + E_{Z,QD} \hat{S}_{z,QD}$$
$$= \frac{U}{2}(\hat{n}_{QD} - \nu)^2 + E_{Z,QD} \hat{S}_{z,QD} + \text{const.},$$

$$H_{SC}^{(\beta)} = \sum_{i,\sigma}^{N} \varepsilon_i c_{i,\sigma,\beta}^\dagger c_{i,\sigma,\beta} - \alpha_\beta d \sum_{i,j}^{N} c_{i,\uparrow,\beta}^\dagger c_{i,\downarrow,\beta}^\dagger c_{j,\downarrow,\beta} c_{j,\uparrow,\beta}$$
$$+ E_c^{(\beta)} \left( \hat{n}_{SC}^{(\beta)} - n_0^{(\beta)} \right)^2 + E_Z^{(\beta)} \hat{S}_z^{(\beta)},$$

$$H_{hyb}^{(\beta)} = \left( v_\beta / \sqrt{N} \right) \sum_{i,\sigma}^{N} \left( c_{i,\sigma,\beta}^\dagger d_\sigma + d_\sigma^\dagger c_{i,\sigma,\beta} \right)$$

Here $\varepsilon_{QD}$ is the energy level and $U$ the electron-electron repulsion on the QD. The QD term can be rewritten in terms of $\nu = 1/2 - \varepsilon_{QD}/U$, the QD level in units of electron number. $d_\sigma$ and $c_{i,\sigma,\beta}$ are the annihilation operators corresponding to the QD and the two SIs labeled by $\beta = L, R$ (left and right). The spin index is $\sigma = \uparrow, \downarrow$. The $N$ SI energy levels $\varepsilon_i$ are spaced by a constant separation $d = 2D/N$, so that $\varepsilon_i = i(2D/N)$ for $i = 1, 2, \ldots, N$. $2D$ is the bandwidth. The levels are coupled all-to-all by a pairing interaction with strength $\alpha$. It generates a superconducting gap

equivalent to the BCS value in the thermodynamic ($N \to \infty$) limit[29], $\Delta \propto 1/\sinh(1/\alpha)$.

The number operators are $\hat{n}_{QD} = \sum_\sigma d_\sigma^\dagger d_\sigma$ for the QD, and $\hat{n}_{SC}^{(\beta)} = \sum_{i=1,\sigma}^{N} c_{i,\sigma,\beta}^\dagger c_{i,\sigma,\beta}$ for each SI; spin operators are $\hat{S}_{z,QD} = (1/2)(d_\uparrow^\dagger d_\uparrow - d_\downarrow^\dagger d_\downarrow)$, and similarly for $\hat{S}_z^{(\beta)}$. $E_c^{(\beta)}$ are the charging energies, with $n_0^{(\beta)}$ the optimal occupation of the SI in units of electron charge. The SIs are coupled to the QD with the hybridisation strengths $\Gamma_\beta = \pi \rho v_\beta^2$, where $\rho = 1/2D$ is the normal-state density of states in each bath. We consider completely symmetrical channels where we drop the label $\beta$, so that $\Gamma = \Gamma_L = \Gamma_R$, $E_c = E_c^{(L)} = E_c^{(R)}$, etc. We take $D = 1$ as the unit of energy.

The results were obtained for $N = 100$ levels in each SI and we set $\alpha = 0.4$, which in the absence of the QD gives $\Delta = 0.16$. This value is chosen so that an appropriate number of levels is engaged in the pairing interaction thus minimizing finite-size effects, while also minimizing the finite-bandwidth effect. The calculations were performed using the density matrix renormalization group method[53] using the iTensor library[54]. The maximal bond dimension in our calculations was 2000, and we truncate singular values smaller than $10^{-10}$. However, we have noticed that reducing the bond dimension by an order of magnitude does not change the results noticeably.

The conserved quantum numbers are the total number of electrons $n$ and the $z$-component of total spin $S_z$. The doublet → singlet excitation energy shown in Fig. 1 is thus given by the energy difference between the ground states of the relevant singlet and doublet sectors $\delta E = E(n = 204, S_z = 0) - E(n = 203, S_z = 1/2)$.

The end-to-end spin correlations shown in Supplementary information Fig. S9 were obtained by extending the model into an alternating chain. Each SI is represented by a single level, $N = 1$.

### Device fabrication

A 120-nm wide InAs nanowire with a 7-nm in-situ grown epitaxial Al shell covering three of its facets was deposited with a micromanipulator on a Si/SiO₂ substrate used as a backgate. The device was defined by a series of electron-beam lithography steps. The Al was patterned into two ≈ 300-nm long islands by Transene-D etching. The nanowire was contacted by Ti/Au (5/200 nm) leads following a gentle argon milling to remove the nanowire native oxide. A 5-nm thick layer of HfO₂ was deposited over the device to insulate it from seven Ti/Au top gates deposited thereafter. Gates 1 and 7 were respectively short-circuited to gates $V_L$ and $V_R$.

### Measurements

All measurements where performed in an Oxford Triton dilution refrigerator at 30 mK. $G$ was measured by biasing the source with a lock-in voltage of 5 $\mu$V at a frequency 84.29 Hz on top of $V_{sd}$, and recording the lock-in current at the grounded drain. Zero-bias $G$ was measured at -18 $\mu$V to account for an offset in the current amplifier. $B$ was aligned with the nanowire axis to maximize the critical field, $B_c$. $B_c$ was estimated at >1.5 T. A single QD was achieved by setting $V_3$, $V_5$ to negative values. To achieve left-right symmetry, QD shells with approximately left-right symmetric binding energy were further fine-tuned with $V_L$ and $V_R$ until $E_{cL}/\Delta_L \approx E_{cR}/\Delta_R$. To achieve the electron-hole symmetric filling of the QD in Fig. 2, $V_N$ was fine-tuned until the bottom left and top right parts of the stability diagram were symmetric. Tuning of $E_{cL}/\Delta_L$ and $E_{cR}/\Delta_R$ was achieved by using two auxiliary QDs, one each to the left and right of the left and right SIs. $E_{cL,R}$ was reduced when these QDs were put in resonance with the drain and source leads. Though in reality a five element QD-SI-QD-SI-QD chain, the device behaved as a shorter SI-QD-SI chain as intended with the outer QDs set in cotunnelling. We speculate that this was due to low tunnel couplings between the auxiliary QDs and the SIs, and/or due to the auxiliary QDs having even occupation.

## Data availability

Data shown in the paper is available on Zenodo at https://doi.org/10.5281/zenodo.10841053.

## Code availability

Code for solving problems with superconducting islands is hosted on the Github repository https://github.com/rokzitko/tensor and the current release is also available on Zenodo[55] at https://doi.org/10.5281/zenodo.10804271.

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

## Acknowledgements
We thank Felix Von Oppen, Ramón Aguado, Martin Žonda, András Pályi and Jens Paaske for useful discussions and Peter Krogstrup for providing nanowire materials. The project received funding from the European Union's Horizon 2020 research and innovation program under the Marie Sklodowska-Curie grant agreement No. 832645 (J. C. E. S., J. N.), QuantERA 'SuperTop' (NN 127900) (J. N.) and FETOpen AndQC (828948) (K. G.-R., J. N.). We additionally acknowledge financial support from the SolidQ project of the Novo Nordisk Foundation (J. C. E. S., J. N.), the Carlsberg Foundation (J. N.), the Independent Research Fund Denmark (J. N.), the Danish National Research Foundation (DNRF 101) (K. G.-R., J. N.), Villum Foundation project No. 25310 (K. G.-R.), KU SCIENCE Visiting Scholar program (R. Ž), the Sino-Danish Center. L. P. and R. Ž. acknowledge the support from the Slovenian Research and Innovation Agency (ARIS) under Grants No. P1-0044, P1-0416, and J1-3008 (L. P. and R. Ž.).

## Author contributions
J.C.E.S. conceived the experiments with input from all co-authors. J.C.E.S. and A.V. did the device fabrication and measurements. J.C.E.S, A.V., K.G.R. and J.N. performed the experimental data analysis. L.P. and R.Ž. did the theoretical modelling. All authors contributed to data interpretation and writing the manuscript.

## Competing interests
The authors declare no competing interests.
