## [Peer Review File · Nature Communications]

Correlation between two distant quasiparticles in separate superconducting islands mediated by a single spinREVIEWER COMMENTS

Reviewer #1 (Remarks to the Author):

The authors investigate a system which is a chain of alternating superconducting islands (SI) and quantum dots (QD). The main point seems to be that a coupling of each QD to two neighbouring SIs leads to an overscreened state with $S=1/2$ rather than a screened one with $S=0$, so that local $S=1/2$ degrees of freedom emerge, which are then coupled in a longer chain. I can only comment on the theoretical part of the paper.

The results are interesting, but the presentation of the paper is very confusing and difficult to follow. It clearly needs much more polishing before being publishable.

Initial motivating statements mention "long-range quantum entanglement" and cite a paper on quantum spin liquids (QSL) as Ref. [1]. Long-range entanglement in QSLs is referring to the constant offset in the entanglement entropy that comes from topological order. This is of course not what is done in the paper, where the authors seem to use "entanglement" synonymously to "correlations". To avoid misleading the reader, I would therefore suggest replacing "entanglement" by "correlations" in the initial section, as well as in the title.

It is not always clear what has been computed in which model. As I understand, the authors use three models:

- (i) 2 SI (1 level) + 1 QD
- (ii) 2 SI (100 levels) + 1 QD
- (iii) many SIs (single-level) + many QDs

In l.233, it says that Fig. 2c,d are obtained with model (ii), but in the inset of Fig. 2 and on l.263, it says that Fig. 2c is obtained with model (i).

- l.72: The mention of "long-range correlations" is confusing in the context of a model with just 2 SI and 1 QD (a trimer has no long range).
- l.250: Γ_β is defined, but is used without the index β in the rest of the paper without further comment.
- l.254: Δ is not defined. I guess this refers to the mean-field/BCS gap, which I think should be $1/\sinh(1/0.4)=0.17?$
- l.248: It should be d_{\uparrow} in the expression for S_z .

In Fig. 2, it is said that a squiggle represents a singlet, but then there can be no squiggles between three spins because of the monogamy of entanglement. So either the squiggle rather means antiferromagnetic correlations or one should use another symbol in a).

Also in Fig.2a, it seems that one arrow for $|1/2, -1\rangle$ should be pointing up?

The term "element of the chain" is only implicitly defined. An explicit definition (sites defined either by an SI or a QD) would help the reader.

Extended Fig. 1. leads to many questions:

Please specify the model for this figure. I guess it's model (iii) from above, for which there is no Hamiltonian given. What is the value of U ? What is the definition of S_i ? (This is also a problem in Fig.1b.) Is the local expectation value subtracted or not?

The correlations in the plot are clearly antiferromagnetic, and describing them as "ferromagnetic correlations between every second element" in the main text (e.g. inset of Fig. 1) is very confusing to me.

It is unclear in what sense the results show long-range order. The total spin is not extensive and the correlations decrease with the distance. It seems rather to be quasi-long-range order, whereby E_c influences the power-law coefficient.

These results beg the question: What is the effective spin model for this case? It looks like an AFM

Heisenberg chain. But then, how can one understand that an odd number of L may result in a singlet state? Why is L chosen to be odd in the first place? Would the ground state of an even- L chain be a singlet?

From the technical point of view, I think that the DMRG calculations for model (ii) are very feasible. Coupled SC grains were in fact already studied in Gobert, Schollwöck, von Delft (2004), for example (though without the QD in between). However, there are no error measures or internal parameters (e.g. bond dimension) given, which does not allow the reader to assess the difficulty of the problem and the accuracy of the results.

Reviewer #2 (Remarks to the Author):

The manuscript under review describes transport experiments conducted in InAs nanowires, where a normal quantum dot is surrounded by two superconducting islands. The device is tuned to a regime where each superconducting island contains a quasiparticle and the dot contains a single electron. In the regime investigated in the experiment, the superconducting islands screen the spin in the QD, resulting in an overscreened state with long-range ferromagnetic spin correlations between the two quasiparticles. I find the topic very interesting, and the geometry new. Also the amount of work is outstanding. However, I think the paper is not ready for publication.

I find the paper extremely hard to read, despite the English is excellent. The main problem is that the manuscript is a mix of experimental results, interpretations, theoretical predictions and claims. The most prominent example is the title: I do not find any experimental evidence of entanglement in this work. Rather, if the model used to describe the data is correct, then long range entanglement would be expected in a longer version of the device. I strongly urge the authors to better separate the theoretical predictions, experimental results, interpretations and outlook. In particular, the title should not contain anything that is not clearly experimentally shown. I would have preferred to see a better separation between theory and experimental figures too.

Both the abstract and the conclusion provide the main message of the paper, but do not say what is the main experimental observation leading to such conclusion. This prevents non-experts from approaching the paper. In particular those not familiar with the physics of YSR states would barely understand what the paper is about. The paper does not explain properly what are the main experimental features to look at. For example, description of Figs.1 d-h is totally absent if not for some hints in the captions.

The vertical label of Figs.3 (c,d) reads "overscreened doublet size". It would be better to give a label a name or symbol that immediately links to the symbols used in Figs. 3 (a,b).

Caption of Fig.1: how are the differences of 8% and 14% obtained?

Line 109 "The ground state is always a doublet, only its nature changes as the coupling becomes large." What is the feature that changes?

Were the results, or parts of it, reproduced in on a second device?

Response to comments of Reviewer #1:

The authors investigate a system which is a chain of alternating superconducting islands (SI) and quantum dots (QD). The main point seems to be that a coupling of each QD to two neighbouring SIs leads to an overscreened state with $S=1/2$ rather than a screened one with $S=0$, so that local $S=1/2$ degrees of freedom emerge, which are then coupled in a longer chain. I can only comment on the theoretical part of the paper.

The results are interesting, but the presentation of the paper is very confusing and difficult to follow. It clearly needs much more polishing before being publishable.

We appreciate that the reviewer finds our results to be interesting while we also admit in hindsight that the story was not easy to follow. In order to make it easier to follow the discussion, we have more clearly separated theoretical and experimental parts of the paper (model first, then experiments), rewritten various parts as well as removed a number of items, including the previous experimental Fig. 3 which is now an Extended Data Figure 3.

Because of the extensive nature of the changes in the manuscript text, with a number of sections entirely rewritten and with significant changes occurring in almost all paragraphs, we have not indicated the changes by colour highlighting (only parts of Fig. 4 text and conclusions are left partially unchanged). We apologise for the inconvenience this may cause.

Initial motivating statements mention "long-range quantum entanglement" and cite a paper on quantum spin liquids (QSL) as Ref. [1]. Long-range entanglement in QSLs is referring to the constant offset in the entanglement entropy that comes from topological order. This is of course not what is done in the paper, where the authors seem to use "entanglement" synonymously to "correlations". To avoid misleading the reader, I would therefore suggest replacing "entanglement" by "correlations" in the initial section, as well as in the title.

We accept this request to clarify the terms. We have thus changed the title to "Correlation between two distant quasiparticles in separate superconducting islands mediated by a single spin" and converted the initial mentions of entanglement into discussions of coupling and correlations. We now also focus solely on the results on correlations in the three element system (two islands and one quantum dot), showing calculations on longer chains only in the extended data.

It is not always clear what has been computed in which model. As I understand, the authors use three models:

(i) 2 SI (1 level) + 1 QD

(ii) 2 SI (100 levels) + 1 QD

(iii) many SIs (single-level) + many QDs

We admit that the introduction of multiple models was confusing. In the main text we use now only one single model, the Richardson + Anderson setup (ii) used in Fig. 1 and Fig. 4, while the simplified picture (i) has been omitted. One extended figure presents result on longer chains (Ext Data Fig. 9), i.e. model (iii), to which we refer only in the final conclusions.

In l.233, it says that Fig. 2c,d are obtained with model (ii), but in the inset of Fig. 2 and on l.263, it says that Fig. 2c is obtained with model (i).

The caption was correct, however, in the new manuscript model (i) is omitted and we focus entirely on model (ii), presented extensively in our new Fig. 1.

- l.72: The mention of "long-range correlations" is confusing in the context of a model with just 2 SI and 1 QD (a trimer has no long range).

Our model is defined in the energy space and hence it indeed does not contain the concept of real space. However, we use it to describe a device of a definite size, with superconducting islands spanning micrometers. With "long-range correlations", we simply meant the existence of correlation on the length scale of the micrometer device size. In the new version we use "long-distance correlations", which has a more appropriate connotation, in the context of the SI-QD-SI device.

- l.250: Gamma_beta is defined, but is used without the index beta in the rest of the paper without further comment.

We have dropped the indices beta and use Gamma as the hybridisation between each island and the QD. This is appropriate, because we only discuss symmetrical devices.

- l.254: Delta is not defined. I guess this refers to the mean-field/BCS gap, which I think should be $1/\sinh(1/0.4)=0.17?$

Yes. This is now clarified in the Methods section.

- l.248: It should be $d_{\{up\}}$ in the expression for S_z .
Corrected.

In Fig. 2, it is said that a squiggle represents a singlet, but then there can be no squiggles between three spins because of the monogamy of entanglement. So either the squiggle rather means antiferromagnetic correlations or one should use another symbol in a).

Squiggles represent some degree of entanglement, possible partial, not necessarily singlets (i.e. maximally entangled spins, which can indeed only be pairwise). The current version of the text makes no reference to singlets.

Also in Fig.2a, it seems that one arrow for $|1/2, -1\rangle$ should be pointing up?

These sketches have been removed for simplicity.

The term "element of the chain" is only implicitly defined. An explicit definition (sites defined either by an SI or a QD) would help the reader.

We have clarified this in the new version of the text.

Extended Fig. 1. leads to many questions:

This is mostly Extended Data Figure 9 in the new version.

Please specify the model for this figure. I guess it's model (iii) from above, for which there is no Hamiltonian given. What is the value of U ? What is the definition of S_i ? (This is also a problem in Fig.1b.)

The model used for these calculations is model (iii), one level representing each SC. This, and the parameter values used, is now stated in the Methods section.

Is the local expectation value subtracted or not?

We plot the expectation value $\langle \mathbf{S}_1 \cdot \mathbf{S}_l \rangle$, where the spin operators are vector quantities. We did not subtract local expectation values.

The correlations in the plot are clearly antiferromagnetic, and describing them as "ferromagnetic correlations between every second element" in the main text (e.g. inset of Fig. 1) is very confusing to me.

Agreed. This statement has been removed.

It is unclear in what sense the results show long-range order. The total spin is not extensive and the correlations decrease with the distance. It seems rather to be quasi-long-range order, whereby E_c influences the power-law coefficient.

Indeed, the correct statement is that there is quasi-long-range order. This has been corrected.

These results beg the question: What is the effective spin model for this case? It looks like an AFM Heisenberg chain. But then, how can one understand that an odd number of L may result in a singlet state?

In the subspace where the number of local moments is equal to the number of subsystems (dots or islands), i.e., with one local moment per "site", the appropriate effective spin model is indeed a Heisenberg chain. In this case, the exchange parameter J can be obtained by a Schrieffer–Wolff transformation for each QD. Such a chain cannot form a singlet state.

However, a SC-QD-SC-.. chain can also be filled with an even number of electrons, and the corresponding lowest energy state is then a singlet. The Hamiltonian projected to the even-parity subspace would be somewhat similar to the tJ model doped with one hole, in a system with an odd number of sites. The lowest-energy state in this subspace is expected to be a singlet.

Why is L chosen to be odd in the first place? Would the ground state of an even- L chain be a singlet?

We choose odd L precisely to demonstrate the emergence of the overscreened state (for the SC-QD-SC case), and its extension resulting in sizeable correlations between the SIs at each end of the chain.

In the case of even L , the chain starts with a SI and ends with a QD (or the opposite), and the end-to-end correlations are antiferromagnetic. The ground state would indeed be a singlet.

From the technical point of view, I think that the DMRG calculations for model (ii) are very feasible. Coupled SC grains were in fact already studied in Gobert, Schollwvov Delft (2004), for example (though without the QD in between). However, there are no error measures or internal parameters (e.g. bond dimension) given, which does not allow the reader to assess the difficulty of the problem and the accuracy of the results.

The method parameters are now given in the Methods section. We also comment on the accuracy of the results (which are highly converged).

The Gobert et al. work is a very different method (DMRG for pairs) applied to a very different problem (two nano grains, with no pair-breaking quantum dot with a local moment, effective charging energy coming from finite (large) level spacing), addressing different physical effects. It is the presence of the QD which makes the problem difficult, since it brings into play the Kondo screening.

Response to comments of Reviewer #2:

The manuscript under review describes transport experiments conducted in InAs nanowires, where a normal quantum dot is surrounded by two superconducting islands. The device is tuned to a regime where each superconducting island contains a quasiparticle and the dot contains a single electron. In the regime investigated in the experiment, the superconducting islands screen the spin in the QD, resulting in an overscreened state with long-range ferromagnetic spin correlations between the two quasiparticles. I find the topic very interesting, and the geometry new. Also the amount of work is outstanding. However, I think the paper is not ready for publication.

We are delighted that the reviewer finds that the investigated system is new and interesting while we also acknowledge the feedback that the experimental presentation was difficult to follow. We have now made an effort to address all issues pointed out by the reviewers, hopefully providing a clearer and more focused presentation of the results. We fully acknowledge the reviewer's comment below that the main observations should be pointed out more clearly.

Because of the extensive nature of the changes in the manuscript text, with a number of sections entirely rewritten and with significant changes occurring in almost all paragraphs, we have not indicated the changes by colour highlighting (only parts of Fig. 4 text and conclusions are left partially unchanged). We apologise for the inconvenience this may cause.

I find the paper extremely hard to read, despite the English is excellent. The main problem is that the manuscript is a mix of experimental results, interpretations, theoretical predictions and claims. The most prominent example is the title: I do not find any experimental evidence of entanglement in this work. Rather, if the model used to describe the data is correct, then long range entanglement would be expected in a longer version of the device.

Prompted by the (justified) comments from both reviewers, we have omitted the term entanglement from title and most paragraphs. We now phrase our statements in terms of correlations and couplings that can indeed be extracted from our modelling and experiments that are also more clearly separated now. The new title is "Correlation between two distant quasiparticles in separate superconducting islands mediated by a single spin"; here, "distant" refers to micron-scale separation of the superconducting islands.

I strongly urge the authors to better separate the theoretical predictions, experimental results, interpretations and outlook. In particular, the title should not contain anything that is not clearly experimentally shown. I would have preferred to see a better separation between theory and experimental figures too.

We have implemented a clearer separation between theory and experimental results. Fig. 1 is dedicated to the theoretical motivation and modelling. Figs. 2-3 are now purely experimental. Only Fig. 4 on magnetic field behaviour compares experimental analysis directly to the model predictions, in separate panels. The discussion of longer ranges/chains is included only in the outlook where we refer to the relevant extended data figure.

Both the abstract and the conclusion provide the main message of the paper, but do not say what is the main experimental observation leading to such conclusion. This prevents non-experts from approaching

the paper. In particular those not familiar with the physics of YSR states would barely understand what the paper is about. The paper does not explain properly what are the main experimental features to look at. For example, description of Figs.1 d-h is totally absent if not for some hints in the captions.

We thank the Reviewer for this comment. We have considered it seriously, which has prompted us to significantly revise the manuscript, including the reorganisation of Figures and the order of discussion. The revised Fig. 1 introduces the system and shows model plots with the evolution of the relevant states as a function of (experimental) parameters, as well as how the overscreened doublet is predicted to appear in the following experimental stability diagrams. Hopefully, this enables the reader to appreciate the experimental evidence for the emerging overscreened doublet state.

We provide in the new version a more explicit discussion of the experimental data, focusing on the simpler manifestations of the states, and we point out the main observations.

The previous panels Figs. 1 d-h have now been made into separate figures (Fig. 2-3) which are now discussed thoroughly in the main text, explaining firstly the tunability and L-R symmetry of the devices (Fig. 2), and the charge patterns and the appearance of the doublet state when tuning the control parameter (Fig. 3). We hope that this provides a clearer progression in the experimental results.

The vertical label of Figs.3 (c,d) reads "overscreened doublet size". It would be better to give a label a name or symbol that immediately links to the symbols used in Figs. 3 (a,b).

The data from Fig. 3 have been removed from the main manuscript in order to simplify the presentation. We do show the results (Extended Data Fig. 3) and the axes labels in c,d have been modified to the appropriate symbol used in panel b.

Caption of Fig.1: how are the differences of 8% and 14% obtained?

The differences between left and right islands were obtained from analyses of the charge stability diagrams, now in Extended Data Fig. 3. This is now explained explicitly in the caption to Fig. 2 where the left-right symmetry is discussed in the new version.

Line 109 "The ground state is always a doublet, only its nature changes as the coupling becomes large." What is the feature that changes?

This figure (Fig. 3) and the related text is omitted from the new version. What we meant is that the GS is a doublet, but at $\Gamma \rightarrow 0$ it has a large contribution of the decoupled doublet (0,1,0). With increasing Γ , the contribution of the overscreened (1,1,1) state grows.

Were the results, or parts of it, reproduced in on a second device?

We currently do not have any further results from another device. We note, however, that there is some degree of robustness in the sense that the same phenomenon was observed over a number of different shells (electron fillings).

REVIEWERS' COMMENTS

Reviewer #1 (Remarks to the Author):

The authors have greatly improved the presentation of the paper. The motivation, story and the theoretical description are much better understandable now. I cannot comment on the experimental challenges or the impact of the result, but the theoretical section is sound and agreement with the experiment regarding the 3 LMs is demonstrated. Overall, I can recommend the paper for publication.

There are still some mistakes in the manuscript:

- p.5: The squares belong inside the brackets: $\langle S_R^2 \rangle$ etc.
- Extended Data Figure 9 is not referenced in the main text
- p.7: vision -> version?

Some more cosmetic improvements:

- p. 29: onedimensional -> one-dimensional
- p.8: I think one should explicitly define ϵ_i , i.e. $\epsilon_i = 2D/N^i$ ($i=1,2,3,\dots$)
- p.9: the "cutoff" is probably the "truncated weight"?
- p.17, Fig. 1: There is enough space to upscale the legend in subfigure g), otherwise it's quite hard to read

Reviewer #2 (Remarks to the Author):

The new version of the manuscript does a much better job in conveying the theoretical predictions, experimental results, and interpretations. The new figures are clearer and easy to follow. I suggest publication in the present form.

Reviewer #3 (Remarks to the Author):

In a combination of experiments and theoretical works, the manuscript by Estarada Saldena and coworkers reports on the realization of a system of three exchange-correlated local magnetic spin moments in the geometry of a quantum dot (QD) connected to two superconducting islands (SIs). Single electron charging effects in both superconducting islands and in the QD allow tuning the system to a state in which the ground state of each individual element of the chain has odd occupancy, and thus carries a single local spin moment. The exchange interaction between the spins is known to lead to complex many-body ground states of the hybrid system. Usually this results in a singlet state shared between the QD and one lead. In the present work however, overscreening of the QD spin from two channels (from the two SIs) allows achieving a new ground state, in which, in spite of their large spatial separation, the spins in the two SIs are correlated. This is a very interesting and timely study, with convincing agreement between experiments and theory. The physics at play is intricate but the previous reviewing round has already allowed significantly improving the manuscript's quality. I recommend acceptance at Nature Communications and append two suggestions for further improvements:

- The end of the first paragraph (after abstract) introduces the two-channel overscreening effect of a QD spin in the presence of superconducting leads. Here the claims of the authors are not properly referenced: are the results (such as the insensitivity of the overscreening effect to the condition of equal couplings to both channels, as in the normal case) known from the literature or

a new result? In any case, it would be satisfactory to understand qualitatively why this crucial condition for overscreening (making this effect rather exotic in normal structures) is lifted with superconducting leads.

- I suggest stating explicitly (before the Methods) that the in-plane magnetic field of 100-120 mT applied does not affect superconductivity. Without this information, one can be tempted to interpret the downturn of the excitation energies in Fig.4e in terms of a weakening of the gap.

Response to comments of Reviewer #1:

The authors have greatly improved the presentation of the paper. The motivation, story and the theoretical description are much better understandable now. I cannot comment on the experimental challenges or the impact of the result, but the theoretical section is sound and agreement with the experiment regarding the 3 LMs is demonstrated. Overall, I can recommend the paper for publication.

We are delighted that the paper is recommended publication and we thank reviewer 1 for the extensive, detailed comments to the previous version that clearly improved our presentation.

There are still some mistakes in the manuscript:

- p.5: *The squares belong inside the brackets: $\langle S_R^2 \rangle$ etc.*

This mistake has been corrected now.

- *Extended Data Figure 9 is not referenced in the main text*

The figure was actually referenced in the conclusions p. 7, “extend throughout the whole chain, see Extended Data Fig. 9”, and also in the Methods section.

- *p.7: vision -> version?*

We have rephrased the sentence to “For example, a longer odd-length chain can be used to demonstrate the self-similarity of the two-channel Kondo state on a tight-binding chain”.

Some more cosmetic improvements:

- *p. 29: onedimensional -> one-dimensional*

Corrected.

- *p.8: I think one should explicitly define ϵ_i , i.e. $\epsilon_i = 2D/N \cdot i$ ($i=1,2,3,\dots$)*

We have expanded this definition.

- *p.9: the "cutoff" is probably the "truncated weight"?*

We agree. We now write “we truncate singular values smaller than ...”

- *p.17, Fig. 1: There is enough space to upscale the legend in subfigure g), otherwise it's quite hard to read*

It was indeed too small. We have increased the font size.

Response to comments of Reviewer #2:

The new version of the manuscript does a much better job in conveying the theoretical predictions, experimental results, and interpretations. The new figures are clearer and easy to follow. I suggest publication in the present form.

We are grateful that the reviewer appreciated our revised manuscript.

Response to comments of Reviewer #3:

In a combination of experiments and theoretical works, the manuscript by Estarada Saldena and coworkers reports on the realization of a system of three exchange-correlated local magnetic spin moments in the geometry of a quantum dot (QD) connected to two superconducting islands (SIs). Single electron charging effects in both superconducting islands and in the QD allow tuning the system to a state in which the ground state of each individual element of the chain has odd occupancy, and thus carries a single local spin moment. The exchange interaction between the spins is known to lead to complex many-body ground states of the hybrid system. Usually this results in a singlet state shared between the QD and one lead. In the present work however, overscreening of the QD spin from two channels (from the two SIs) allows achieving a new ground state, in which, in spite of their large spatial separation, the spins in the two SIs are correlated. This is a very interesting and timely study, with convincing agreement between experiments and theory. The physics at play is intricate but the previous reviewing round has already allowed significantly improving the manuscript's quality. I recommend acceptance at Nature Communications and append two suggestions for further improvements:

We agree with this summary and acknowledge the additional suggestions below.

- The end of the first paragraph (after abstract) introduces the two-channel overscreening effect of a QD spin in the presence of superconducting leads. Here the claims of the authors are not properly referenced: are the results (such as the insensitivity of the overscreening effect to the condition of equal couplings to both channels, as in the normal case) known from the literature or a new result? In any case, it would be satisfactory to understand qualitatively why this crucial condition for overscreening (making this effect rather exotic in normal structures) is lifted with superconducting leads.

Prompted by these comments, we now discuss the conditions in more detail and have added some references and the sentences in bold to the introduction:

“...leading to a frustrated doublet state where the impurity is completely screened, but a many-body spin-1/2 remains, smeared across the system. **This is an unstable fixed point of the renormalization group flow which only exists for symmetric coupling to both leads. In the case of an asymmetry, the system flows towards the state where the screening comes completely from the more strongly coupled channel [Cox1998]. For this reason, difficult fine-tuning is required to demonstrate the overscreened state in QD devices [Oreg2003,Potok2007].** Overscreening is energetically disfavored in the superconducting (SC) case as it requires the presence of two additional finite-energy quasiparticles. However, SIs with a large charging energy can be tuned into an odd-occupancy regime where each SI contains one lone quasiparticle and here a superconducting version of overscreening emerges. **In superconducting systems, the Kondo renormalization process is cut off at the energy scale of the gap [Fabrizio2017], and overscreening is expected to exist** even if the couplings to both SIs are not strictly the same, i.e., the device does not need to be perfectly mirror (left-right) symmetric. **The theoretical model that we present indeed anticipates such a state and furthermore shows that the screening of the QD comes from two quasiparticles: ...”**

Furthermore, in modelling subsection we have added:

“It is worth commenting on the differences between Kondo-type Hamiltonians with no charge transfer processes, and more realistic Anderson-type Hamiltonians that more adequately model quantum dots. Even in the presence of charging energy on SIs, the Anderson model does not behave in the same way as the Kondo model. This is due to the hybridisation, which leads to the formation of bonding and

antibonding orbitals, as discussed above. For this reason, the physics of the overscreened state in the Kondo case [Fabrizio2017] is different from that of the overscreened state in the Anderson case discussed in this work. In particular, there is no self-dual point and no universality of the energy of the subgap state.”

- I suggest stating explicitly (before the Methods) that the in-plane magnetic field of 100-120 mT applied does not affect superconductivity. Without this information, one can be tempted to interpret the downturn of the excitation energies in Fig.4e in terms of a weakening of the gap.

In addition to the statement on the magnetic field in Methods, we have now added a sentence when introducing Fig. 4: “The applied magnetic field is parallel to the nanowire axis and much weaker than the in-plane critical field of the superconducting shell.”, and we have omitted “large” from the sentence “Further evidence for the presence of the OS state is obtained by using a magnetic field ...”. Thus it should now be clear that the field does not affect the bulk superconductivity of the system.